# From Random to Relevant: Harnessing Salient Masks in Non-IID Efficient Federated Learning

## Abstract

Federated learning (FL) offers the ability to train models using decentralized data at client sites, ensuring data privacy by eliminating the need for data centralization. A predominant challenge with FL is the constrained computation and narrow communication bandwidth, particularly evident in resource-restricted edge client nodes. Various solutions, such as transmitting sparse models and iterative pruning have been suggested to tackle this. However, many existing methods necessitate the transmission of full model weights throughout the training, rely heavily on arbitrary or random pruning criteria or costly iterative pruning schedules.

In this work, we propose SSFL, a streamlined approach for sparse decentralized FL training and communication. SSFL identifies a subnetwork prior to training, leveraging parameter saliency scores keeping in mind the distribution of local client data in non-IID scenarios. Distinctively, only the sparse model weights are communicated in each round between client models in a decentralized manner, sidestepping the conventional need of transferring the complete dense model at any phase of training. We validate SSFL's effectiveness using standard non-IID benchmarks, noting marked improvements in the sparsity-accuracy trade-offs. Finally, we deploy our method in a real-world federated learning framework and report improvement in communication time.

## 1 Introduction

The explosion of deep learning over the last decade has completely revolutionized entire fields including computer vision, natural language processing, recommendation systems, and others. In recent times, deep learning models have continued to grow in size, and with it distributed and collaborative training of such models in parallel has become a requirement. In many applications such as Internet of Things (IOT) and healthcare, it is often the case that sensitive data is distributed in sites over great physical distance and a model needs to be trained that learns on this distributed data. It is also of paramount importance that such models are trained to preserve the privacy of the data without sharing it. In many applications, data is aggregated from various organizations or devices and are pooled in a central server or a platform to train a model which is then dispersed in local sites. This becomes problematic when the data contains sensitive information. For example, to develop a neuroimaging model a series of different hospitals might want to share their data to collaboratively train the model. However, sharing patient information with a central server can reveal sensitive information and raises broad ethical concerns. A relatively recent scheme of training that tackles this setting is federated learning (FL). Federated learning is a collaborative learning technique where different devices or organizations train local models and share training information instead of sharing their data.

Federated learning is an emerging distributed learning paradigm for decentralized data that aims to address many of the above issues including privacy McMahan & Ramage (2017). Federated learning allows decentralized local sites to collaboratively train a shared model without sharing their local data. In the FL paradigm, a central server coordinates the training process and each participating client site (or device) communicates only the model parameters keeping the local data private. However, in many domains and applications, the data generated might be highly heterogeneous and non-IID (independent and identically distributed) Zhu et al. (2021). Moreover, in many scenarios,

the communication and computational resources are often limited in client-edge devices. Therefore, the three most pertinent challenges in the application of FL are statistical heterogeneity of the data, communication bandwidth, and computational cost Kairouz et al. (2021); Li et al. (2020b). In this work, we aim to address the challenges of communication efficiency and computational cost in the decentralized FL setting.

**Federated Learning** In the general federated learning (FL) setting, a central server tries to find a global statistical model by periodically communicating with a set of clients that solve the following problem as formulated by McMahan et al. (2017); Konečný et al. (2016); Bonawitz et al. (2019):

$$\min_{w \in \mathbb{R}^d} f(w) \quad \text{where} \quad f(w) = \frac{1}{n} \sum_{n=1}^{N} f_i(w). \tag{1}$$

Typically, the objective function at each client is taken as $f_i(w) = \ell(x_i, y_i; w)$, which is the loss of the network prediction on a sample $(x_i, y_i)$. We assume that the data is partitioned over a total of $K$ clients, with $\mathcal{P}_k$ the set of indices of the samples on client $k$ and $n_k = |\mathcal{P}_k|$. Thus, the objective in equation 1 can be re-written as:

$$f(w) = \sum_{k=1}^{K} \frac{n_k}{n} F_k(w) \quad \text{where} \quad F_k(w) = \frac{1}{n_k} \sum_{i \in \mathcal{P}_k} f_i(w). \tag{2}$$

In the typical distributed optimization setting, the IID assumption is made, which says the following: if the partition $\mathcal{P}_k$ was created by distributing the training data over the set of clients uniformly at random, then we would have $\mathbb{E}_{\mathcal{P}_k}[F_k(w)] = f(w)$, where the expectation is over the set of examples assigned to a fixed client $k$. In this work, we consider the non-IID setting where this does not hold and $F_k$ could be an arbitrarily bad approximation to $f$.

When designing an FL training paradigm a set of core considerations have to be made to maintain data privacy, and address *statistical* or *objective* heterogeneity due to the differences in client data, and resource constraints at the client sites. A range of work tries to address the issue of heterogeneous non-IID data McMahan et al. (2016); Kulkarni et al. (2020), however, many research also suggest that deterioration in accuracy in the FL non-IID setting is almost inevitable Zhao et al. (2018). In recent times, with the goal of efficient FL, the effort is also being made to reduce the communication cost Chen et al. (2019); Mills et al. (2019); Xu et al. (2020).

**Neural Network Pruning** Like most areas in deep learning, model pruning has a rich history and is mostly considered to have been explored first in the 90's Janowsky (1989); LeCun et al. (1990); Reed (1993). The central aim of *model pruning* is to find subnetworks within larger architectures by removing connections. Model pruning is very attractive for a number of reasons, especially for real-time applications on resource-constraint edge devices which is often the case in FL and collaborative learning. Pruning large networks can significantly reduce the demands of inference Elsen et al. (2020) or hardware designed to exploit sparsity Cerebras (2019); Pool et al. (2021). More recently the *lottery ticket hypothesis* was proposed which predicts the existence of subnetworks of initializations within dense networks, which when trained in isolation from scratch can match in accuracy of a fully trained dense network. This rejuvenated the field of sparse deep learning Renda et al. (2020); Chen et al. (2020) and more recently the interest spilled over into sparse reinforcement learning (RL) as well Arnob et al. (2021); Sokar et al. (2021). Pruning in deep learning can broadly be classified into three categories: techniques that induce sparsity before training and at initialization Lee et al. (2018); Wang et al. (2020); Tanaka et al. (2020), during training Zhu & Gupta (2018); Ma et al. (2019); Yang et al. (2019); Ohib et al. (2022) and post-training Han et al. (2015); Frankle et al. (2021). Among these for Federated Learning applications pruning at initialization holds the most promise due to the selection of a subnetwork right at the start of training and the potential to only train a subset of the parameters throughout the whole training process.

**Sparsity and Pruning in Federated Learning** Federated Learning (FL) has evolved significantly since its inception, introducing a myriad of techniques to enhance efficiency, communication, and

computation. Interest in efficiency has increased further with the proposal of the Lottery Ticket Hypothesis (LTH) by Frankle & Carbin (2019). This hypothesis suggests that within large networks lie hidden *lottery tickets* or smaller, efficient sub-networks that, when isolated, demonstrate impressive performance. However, pinpointing these tickets traditionally can be quite challenging and in the original work, the strategy was to iteratively train and prune which if applied to FL training would make it more inefficient. However, variations of this idea have been used in Li et al. (2020a); Jiang et al. (2022); Seo et al. (2021); Liu et al. (2021); Babakniya et al. (2022) in the FL setting, however many of them either suffer from the same issue of iterative pruning and retraining which is extremely costly or decline in performance when retraining is skipped. Previous studies Lin et al. (2017); Barnes et al. (2020), have investigated variations of the top-k selection in the distributed SGD training framework, with a particular focus on sparse gradients at each training step. We consider these methods to be distributed or parallel training strategies and differentiate them from federated training where data privacy and access from the point of server is a concern and infrequent communication is vital for efficiency.

In the Federated Learning (FL) setting, pruning has been explored with mixed success, often employing a range of heuristics. Works such as Munir et al. (2021) focused on pruning in resource-constrained clients, Yu et al. (2021) on using gated dynamic sparsity, Liu et al. (2021) performs a pre-training on the clients to get mask information, and Jiang et al. (2022) relies on an initial mask selected at a particular client, followed by a FedAvg-like algorithm that performs mask readjustment every $\Delta R$ rounds. However, they either do not leverage sparsity fully for communication efficiency, suffer from performance degradation or is not designed for the challenging, realistic non-IID setting.

Works such as Dai et al. (2022); Bibikar et al. (2022) utilize dynamic sparsity similar to Evci et al. (2020) extending it to the FL regime, however, they both start with random masks as an initialization. A recent focus has also been placed on random masking and optimizing the mask Setayesh et al. (2022) or training the mask itself instead of training the weights at all Isik et al. (2022). Furthermore, in light of the recent surge in the development of large language models (LLMs), the exploration of federated learning in this domain is relatively new Fan et al. (2023); Yu et al. (2023). Investigating sparsity and efficiency within this context presents a fascinating and promising avenue for future research.

**Connection importance in FL**   We did not discover any studies that directly employ connection importance at initialization in the Federated Learning (FL) context. Nevertheless, certain research like Jiang et al. (2022) incorporated gradient-based importance criterion, SNIP Lee et al. (2018); De Jorge et al. (2020), within their comparative benchmarks, but calculating such scores only on the local data of one particular client site in the FL setting, resulting in underperformance. Similarly, Huang et al. (2022) presents connection saliency in their benchmark but assumes that to calculate meaningful parameter importance scores, the server has access to a public dataset on which such saliency scores can be calculated. However, that is in general unlikely to happen and would lead to privacy concerns if such constraints were enforced.

In contrast, in this work, we show that, finding sub-networks using connection saliency taking into account the distributed client training data results in improved performance compared to methods that rely on dynamic sparsity but start from random sparse masks in the non-IID FL regime Dai et al. (2022).

## 2   CONTRIBUTIONS

In this study, we introduce SSFL , a new framework for sparse decentralized federated learning. Our method's main advantage is its ability to train sparse models using information from the data distribution across all client sites. We identify a sparse network or subnetwork to be trained *before* the actual Federated training starts, extending gradient-based saliency scores Mozer & Smolensky (1988); Lee et al. (2018); De Jorge et al. (2020) to the FL setting, taking into account the non-IID nature of the data distribution. We share the mask with all client sites *once* prior to the commencement of training. This approach allows us to transmit only very sparse weights to a subset of clients in each communication round throughout the entire training phase, effectively reducing communication bandwidth. As the server and all client models begin with the same initialization and mask, there is no need to share dense model weights during training, unlike many modern methods. We summarize our contributions as follows:

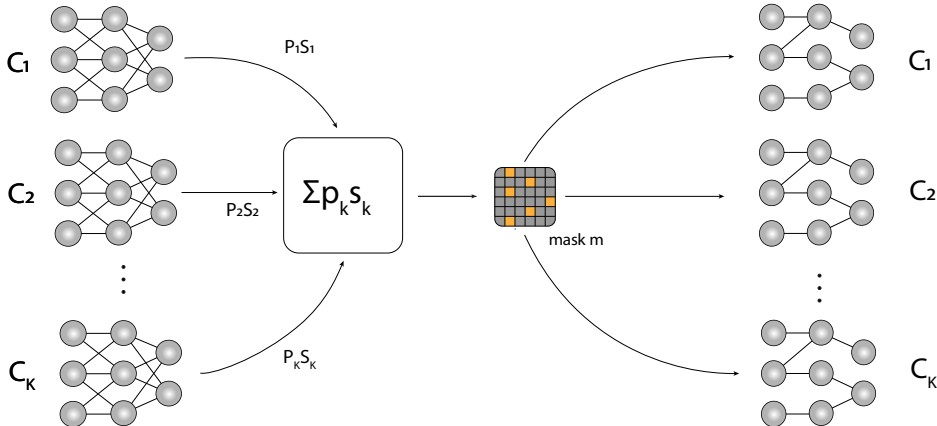

Figure 1: Illustration of the distributed connection importance in the non-IID setting. The parameter saliency scores from each site calculated on local minibatches of equal class distribution are aggregated, weighing them with the proportion of the data available at that site. The common mask generated from that score is applied to local client models.

- We propose a novel sparse Federated Learning paradigm called SSFL to train only a subset of the parameters of client models in a decentralized manner, resulting in a highly communication-efficient federated training technique, resulting in sparse models at client sites.

- We propose a generalized version of gradient-based connection importance criterion for Federated Learning in the non-IID setting.

- SSFL does not need to share dense model parameters or masks during the training phase as we start with the same initialization and only transmit sparse parameters each communication round.

- We avoid adding a range of bells and whistles common in FL works and provide a simple framework that performs well and efficiently and improves upon recent works. We demonstrate our method's efficacy in a range of standard non-IID FL tasks and compare it to existing methods including those that focus on communication efficiency.

- We demonstrate our technique in a real-world federated learning framework that trains neuroimaging models and reports wall-clock time speed up.

## 3 PROPOSED APPROACH

We now introduce our parameter-importance based sparse FL framework Salient Sparse Federated Learning (SSFL) and explain the method. The principle steps in our decentralized federated training mechanism are (1) compute a parameter saliency score based on the decentralized client data distribution (2) find a *common* sparse-network for the local client models based on the decentralized parameter saliency scores (3) train the common sub-network in the client models and communicating only a fraction of the parameters intermittently in each communication round. In subsection 3.1 we elaborate the procedure to find sub-networks before training and in subsection 3.2 we detail our federated training method. In Appendix A.5, we also report communication speed-up in a neuro-imagaging FL system in use currently.

### 3.1 CONNECTION IMPORTANCE

We consider a neural network $f_k$ at site $k$, parametrized by $\boldsymbol{w}_k \in \mathbb{R}^d$ with the parameters $\boldsymbol{w}_{k,0}$ at initialization. The objective is to minimize the empirical risk $\mathcal{L}_k = \frac{1}{M_k} \sum_i \ell(\boldsymbol{w}_{k,0}; \boldsymbol{x}_i, y_i)$ given

a training set $\{(\mathbf{x}_i, y_i)\}_{i=1}^{M_k} \sim \mathcal{D}_k$ the local data distribution at site $k$. A sub-network within this network is defined as a sparse version of this network with a mask $\mathbf{m} \in \{0,1\}^{|\boldsymbol{w}_k|}$ that results in a masked network $f(\boldsymbol{w}_k \odot \mathbf{m}; \boldsymbol{x}_i, y_i)$.

In general, gradient-based connection importance criterions consider the Taylor expansion of the change in the loss to approximate the importance of a neuron or connections. Varying forms of this have been proposed in the literature, the first, probably three decades ago by Mozer & Smolensky (1988) and more recently by Lee et al. (2018); De Jorge et al. (2020) and reformulated in Frankle et al. (2021); Shen et al. (2022) which we describe next. Since the goal is to measure the importance or saliency of each connection an auxiliary indicator variable $\mathbf{c} \in \{0,1\}^d$ is introduced representing the degree of importance of parameter $\mathbf{w}$. Given a sparsity level $\lambda$, the objective can be re-written as follows:

$$\mathcal{L} = \frac{1}{M} \sum_i \ell(\mathbf{c} \odot \boldsymbol{w}_0; \boldsymbol{x}_i, y_i) \tag{3}$$

$$\text{s.t.} \quad \boldsymbol{w} \in \mathbb{R}^d, \quad \mathbf{c} \in \{0,1\}^d, \quad \|c\|_0 \leq \lambda$$

The main goal of the above formulation is to use the $j$-th element $c_j$ of $\mathbf{c}$ as a metric for the importance of each connection $w_j$, where $w_j$ indicates the $j$-th connection, which enables us to determine the saliency of each connection by measuring its effect on the loss function. The core idea of the importance criteria is to preserve the parameters that has maximum impact on the loss under perturbation. The importance criterion is then formulated as:

$$g_j(\boldsymbol{w}; \mathcal{D}) = \left. \frac{\partial \mathcal{L}(\boldsymbol{w} \odot \mathbf{c})}{\partial \mathbf{c}} \right|_{\mathbf{c}=1} = \frac{\partial \mathcal{L}(\boldsymbol{u})}{\partial \boldsymbol{u}} \cdot \frac{\partial \boldsymbol{u}}{\partial \mathbf{c}} = \frac{\partial \mathcal{L}(\boldsymbol{w})}{\partial \boldsymbol{w}} \odot \boldsymbol{w} \tag{4}$$

Where we use a substitution of variable $\boldsymbol{u} = \boldsymbol{w} \odot \mathbf{c}$ which implies, $\frac{\partial \boldsymbol{u}}{\partial \mathbf{c}} = \boldsymbol{w}$ and $\frac{\partial \mathcal{L}(\boldsymbol{u})}{\partial \boldsymbol{u}} = \frac{\partial \mathcal{L}(\boldsymbol{w})}{\partial \boldsymbol{w}}$ at $\mathbf{c} = 1$ and relaxing the binary constraint on the indicator variables $\mathbf{c}$, allowing it to take on real values $c \in [0,1]$. To achieve our primary objective of identifying significant connections towards the start of training, we utilize the magnitude of the gradient $|g_j(\boldsymbol{w}; \mathcal{D})|$ from equation-3 as an indicator of connection importance. A higher value of this magnitude suggests that the corresponding connection $w_j$, associated with $c_j$, has a substantial influence on the loss function, implying its significance in the training process and should be preserved. Consequently, a saliency score for each connection can be calculated based on this principle as follows:

$$s_j = \left| g_j(\boldsymbol{w}; \mathcal{D}) \right| = \left| \frac{\partial \mathcal{L}(\boldsymbol{w})}{\partial \boldsymbol{w}} \odot \boldsymbol{w} \right| \tag{5}$$

We extend the saliency criterion in equation 4 to the Federated Learning setting and use it for computing parameter importance scores that take into account the distributed dataset available across the client sites.

**Parameter Saliency in the FL setting:**   Research by Huang et al. (2022); Jiang et al. (2022) indirectly investigated a variation of the importance criterion in equation 5. When benchmarking their methods against this equation as a baseline, they discovered that its direct application in the Federated Learning (FL) setting did not yield satisfactory results. During benchmarking, they explored it in a centralized setting and computed the scores only with either assuming that a portion of the data is publicly available in the server Huang et al. (2022) or only using it on one particular random client to get the mask Jiang et al. (2022). In contrast, in this work, we generalized the notion of parameter importance in the non-IID FL setting and accumulated the importance scores in a decentralized manner based on the local data distribution. For $K$ different client models we could initially compute $K$ different unique local masks $\{\mathbf{m}_k\}_{k=1}^K$ that uniquely adapts to the local data at the client sites. However, this results in suboptimal performance and we provide an observation explaining this in Appendix-A.2. Hence, our goal is to find a single sub-network with $\mathbf{m}$ that represents the data from all the client nodes. We are sharing the global mask among all the client models once at the start of training and never again throughout the complete federated training process.

In the FL setting for $N$ different client sites with the common initialization $\boldsymbol{w}_0$ and unique data distribution $D_k$ at site $k$, using the saliency criterion in equation 5 we get the saliency score $s_k(\boldsymbol{w}_0; \boldsymbol{x}, y)$ for the $k^{th}$ site where $(\boldsymbol{x}, y) \sim \mathcal{D}_k$. To compute the scores $s_k$ we pass a few minibatches of data and average the saliency scores over the few minibatches at site $k$. We craft the minibatches to have

equal samples from each class of the local dataset $\mathcal{D}_k$, this is to avoid the parameter saliency scores becoming biased towards some classes within the local dataset and disproportionately preserving parameters that are important for a particular class.

To create the global mask $\mathbf{m}$ we average the saliency scores $s_k$ from all sites and get the aggregated global saliency score $s$ from $K$ different sites weighted by the proportion $p_k$ of the dataset available at the local site $k$.

$$s = \sum_{k=1}^{K} p_k s_k(\boldsymbol{w}_0) \tag{6}$$

We then apply the *top-l* operator to find the most important connections for the current initialization and the data on all the client sites. Thus, to generate the global mask $\mathbf{m} \in \{0,1\}^{|\boldsymbol{w}|}$ we select for the *top-l* ranked connections as:

$$\mathbf{m} = \mathbb{1}[s(w_j) \geq s(w_l)] \tag{7}$$

where, $w_l$ is the $l^{th}$ largest parameter in the model and $\mathbb{1}[\cdot]$ is the indicator function.

## 3.2 METHOD DESCRIPTION: SSFL

In our method, a decentralized sparse FL training technique is proposed with the goal to tackle the problems of data heterogeneity when training sparse FL models. We start with a common initialization $\boldsymbol{w}_0$ at all the client models after averaging all $K$ initialized model weights. Next, saliency scores are calculated for each connection in the network based on the data available across the clients according to the equation 4. At this stage, each client has a unique set of saliency scores for connections in the network $f_k$ based on the local data available at that site. All the clients share these scores to each other and a mask $\mathbf{m}$ is created corresponding to the top-$k$ % of the aggregated saliency scores $s = \sum_{k=1}^{K} p_k s_k$ at each local site. This results in each site having the same mask $\mathbf{m}$ and the same initial weight $\boldsymbol{w}_0$. This mask is then used for training the model $f_k(\boldsymbol{w} \odot \mathbf{m}; \boldsymbol{x}, y)$ at site $k$ on their local data $(\boldsymbol{x}, y) \sim \mathcal{D}_k$.

For the decentralized federated training among a total of $K$ clients, the clients are trained locally, and at the end of local training they share their trained parameters among the sites; this is termed as a *communication round*. The models across all the sites are trained in this fashion for $R$ communication rounds. In each communication round $r$, only a random subset $\mathcal{C}' = \{c_1, c_2, ..., c_{K'}\}$ of $K'$ clients where $\mathcal{C}' \subseteq \mathcal{C}$ the set of all clients, and $K' \leq K$ are trained on their local data (a way to avoid the *straggler effect* in the real world, wherein a large client group the update might be bottle-necked by the slowest, most resource-constrained client site). At the end of local training, the client models only share their sparse masked weights $\boldsymbol{w}_m^{\mathcal{C}'} = \boldsymbol{w}_k \odot \mathbf{m}$ among the selected clients in $\mathcal{C}'$ using the compressed sparse row (CSR) encoding. The algorithm for the training process is delineated in Algorithm 1.

## 3.3 DATASET AND NON-IID PARITION

We have evaluated the method on CIFAR-10 and CIFAR-100 Krizhevsky et al. (2009). For simulating non-identical data distributions across the federating clients we use two separate data partition strategies. Class distribution of these samples are illustrated in Appendix A.3

**Dirichlet Partition**   We use Dirichlet (Dir) Partition following works Hsu et al. (2019), where we partition the training data according to a Dirichlet distribution $Dir(\alpha)$ for each client and generate the corresponding test data for each client following the same distribution, similar to work Dai et al. (2022); Bibikar et al. (2022). We specified the $\alpha = 0.3$ for CIFAR10 and $\alpha = 0.2$ for the CIFAR-100.

**Pathological Partition**   Pathological partition of the data is used for partitioning similar to Zhang et al. (2020) where a limited number of classes are only assigned to each client at random from the total number of classes. For our experiments, we restricted each client to possess 2 classes for CIFAR-10 and 10 classes for the CIFAR-100 dataset.

---

**Algorithm 1** SSFL

---

**Input:** Total number of clients $K$; Total communication rounds $R$; Total local training steps $T$ in each communication round; Number of clients $K'$ participating in each round.

**Output:** Sparse local models $\hat{w}_m^{\mathcal{C}}$

1: **for all** clients $k$ in parallel **do**
2:     $w_0 \leftarrow \frac{1}{K}\sum_{k=1}^{K} w_{k,0}$ # Gather all initial $w_{k,0}$ client weights and average in client.
3:     Calculate proportion of data $p_k$ at client $k$. # Each client sends sample counts to all clients
4:     $s \leftarrow \sum_{k=1}^{K} p_k s_k(w_0)$ # Gather all saliency scores at client $k$ and aggregate
5:     $\mathbf{m} = \mathbb{1}[s(w_j) > s(w_\lambda)]$ # calculate common mask from aggregated saliencies
6:     $w_{k,m} \leftarrow w_0 \odot \mathbf{m}$  #apply the mask at site $k$
7: **end for**

8: **for** $r = 0$ to $R-1$ **do**
9:     $c_1, c_2, ..., c_{\hat{K}} \sim \text{Unif}(\mathcal{C})$ # Sample $\hat{K}$ clients uniformly from the set of all clients $\mathcal{C}$
10:     **for** site $k$ in parallel for all $\hat{K}$ clients **do**
11:         $w_m \leftarrow \text{csr}(w_{k,m})$; #Gather all masked weights $w_{k,m}$ where $k \in \{1,2,3,\dots \hat{K}\}$
12:         $w_m^{\hat{\mathcal{C}}} \leftarrow \left(\frac{1}{\hat{K}}\sum_{k=1}^{\hat{K}} w_{k,m}\right)$  #combine the weights of models in the selected sites
13:         $\hat{w}_m^0 \leftarrow w_m^{\hat{\mathcal{C}}}$
14:         **for** $t = 0$ to $T-1$ **do**
15:             $\mathbf{g}_m^t \leftarrow \nabla_w \mathcal{L}(\hat{w}_m^t; x^t, y^t) \odot \mathbf{m}$ # calculate and mask gradients
16:             $\hat{w}_m^{t+1} \leftarrow \hat{w^t}_m - \eta \mathbf{g}_m^t$ # take optimization step with masked gradients on masked weights
17:         **end for**
18:     **end for**
19:     transmit the non-zero elements of global models $\hat{w}_m^{\mathcal{C}}$ back to the clients.
20: **end for**

---

## 4 EXPERIMENTS

We compare SSFL to a range of Federated learning methods and decentralized federated learning methods in this section. The results of the comparison on non-IID CIFAR-10 dataset are presented in Table 1, on the non-IID CIFAR-100 dataset in Table 2 and on the non-IID TinyImageNet data on Table 3. We also include a comparison between the sparse FL method of Dai et al. (2022) and ours at varying levels of sparsity in Fig 2 on CIFAR-10 and CIFAR-100 dataset. Moreover, we also deploy our model in a real world Federated Learning framework and report wall-time improvement in Appendix A.5 and test our method on a complex neuro-imaging structural brain MRI task in Appendix A.6.

**Experiment Details and hyper-parameters:** We use the SGD optimizer for all techniques, employing a weighted decay parameter of 0.0005. With the exception of the "Ditto" method, we maintain a constant of 5 local epochs for all methods. However, for "Ditto," in order to ensure equitable comparison, each client undertakes 3 epochs for training the local model and 2 epochs for training the global model. Our initial learning rate stands at 0.1 and diminishes by a factor of 0.998 post each round of communication similar to Dai et al. (2022). Throughout all experiments, we use a batch size of 16 due to the usage of group normalization Wu & He (2018). We execute $R = 500$ global communication rounds for CIFAR-10, CIFAR-100. For the experiments in Table 1, 2 and 3 we use a sparsity level $s = 50\%$ similar to Dai et al. (2022). Moreover, we present sparsity vs accuracy comparison between SSFL and related sparse FL methods including to the baseline random-masking on non-IID data in Fig-2 with, increasing sparsity levels.

**Baselines.** We compared our method with a range of baselines centralized and decentralized baselines. Centralized baselines include **FedAvg** McMahan et al. (2017), **FedAvg-FT** Cheng et al. (2021), **Ditto** Li et al. (2021), **FOMO** Zhang et al. (2020), and **SubFedAvg** Vahidian et al. (2021). For the decentralized FL setting, we take the sparse Dis-PFL Dai et al. (2022) and the commonly used

| Method | Dir. Part Acc | Path Part Acc | Comms (MB) | Sparse |
|---|---|---|---|---|
| **SSFL** | **88.29%** | **94.61%** | 223.4 | ✓ |
| Dis-FPL | 85.12% | 91.25% | 223.4 | ✓ |
| SubFedAvg | 76.50% | 91.20% | 278.8 | ✓ |
| top-k | 53.33% | - | 223.4 | ✓ |
| Random | 41.61% | - | 223.4 | ✓ |
| Fed-PM | 57.30% | - | - | optimal |
| FedAvg | 86.04% | - | 446.9 | ✗ |
| FedAvg-FT | 88.02% | 92.90% | 446.9 | ✗ |
| D-PSGD-FT | 83.05% | 88.74% | 446.9 | ✗ |
| Ditto | 83.50% | 83.54% | 446.9 | ✗ |
| FOMO | 66.21% | 88.25% | 446.9 | ✗ |

Table 1: Comparison of SSFL with similar decentralized federated and sparse federated learning methods on ResNet18 on the Cifar10 non-IID dataset.

**D-PSGD** Lian et al. (2017) as another method we compare to. Similar to Dai et al. (2022), to accommodate **D-PSGD** to the FL setting, we extend the local training from only one iteration of stochastic gradient descent to several epochs over local data and, similarly to **FedAvg-FT**, we extend it to **D-PSGD-FT** as another baseline where the final models are acquired by performing few fine-tuning steps on the global consensus model with local data. FedPM trains sparse random masks instead of training model weights and finds an optimal sparsity by itself Isik et al. (2022). We use the official implementation of FedPM. For Top-k baseline, the top-k weights from each local model at the end of $T$ steps are shared among the selected clients in each communication round and are aggregated. We differentiate this from top-$k$ gradient sparsification in distributed training Barnes et al. (2020); Lin et al. (2017) . The experimental setup is explained in detail in Appendix A.2.

| Method | Dir. Part Acc | Path Part Acc | Comms (MB) | Sparse |
|---|---|---|---|---|
| **SSFL** | **61.37%** | 52.01 % | 224.0 | ✓ |
| Dis-PFL | 59.21% | 44.74 % | 224.0 | ✓ |
| FedAvg-FT | 59.42% | **52.47 %** | 448.7 | ✗ |
| D-PSGD-FT | 50.27% | 27.58 % | 448.7 | ✗ |
| Ditto | 43.50% | 51.90 % | 448.7 | ✗ |
| FOMO | 32.50% | 45.25 % | 448.7 | ✗ |
| SubFedAvg | 47.25% | 46.04 % | 346.6 | ✓ |

Table 2: Comparison of SSFL with similar decentralized federated and sparse federated learning methods on CIFAR100 non-IID dataset using the ResNet18.

| Method | Dir. Part Acc | Comms (MB) | Sparse |
|---|---|---|---|
| **SSFL** | **19.4%** | 224.0 | ✓ |
| Dis-PFL | 8.27 % | 224.0 | ✓ |
| FedAvg-FT | 18.21% | 448.7 | ✗ |
| D-PSGD-FT | 11.98% | 448.7 | ✗ |
| Ditto | 17.80% | 448.7 | ✗ |
| FOMO | 04.27% | 448.7 | ✗ |
| SubFedAvg | 18.76% | 346.6 | ✓ |

Table 3: Comparison of SSFL with similar decentralized federated and sparse federated learning methods on TinyImagenet non-IID dataset using the ResNet18.

**Comparison with sparse FL baseline with increasing sparsity levels:** We next compare with a range of recent sparse FL training methods including DisPFL Dai et al. (2022), SubFedAvg Vahidian et al. (2021), FedPM Isik et al. (2022) and with random masking with varying sparsity levels.

DisPFL is a sparse FL technique that randomly prunes each layer similar to Evci et al. (2020) and uses the prune and regrow method from that work as well, resulting in a dynamically sparse method. In contrast, instead of randomly choosing the initial mask, we use the parameter saliency criterion utilized in equation 4 and extend it to the distributed setting. We observe in Fig 2 that SSFL consistently performs better than Dis-PFL in a range of sparsities in the selected tasks. This is probably due to a better choice of the initial sparse sub-network using the importance criterion.

Another difference is that, in DisPFL different local clients have different levels of sparsity and a final model averaging is done, where the final model becomes denser due to union of many sparse subnetworks. We however retain the same mask in all the clients and start from the same initialization in all the clients, result in equivalent sparsity in all the clients; this also leaves open the potential of keeping sparse global models in a centralized FL setting.

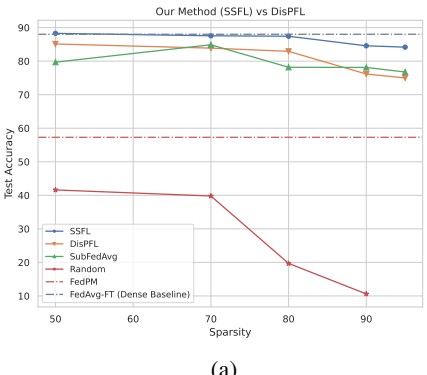
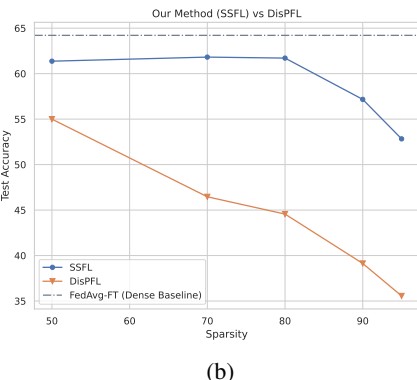

(a)                                            (b)

Figure 2: Sparsity vs accuracy comparison between (a) SSFL, DisPFL, SubFedAvg, FedPm and Random Masking on Cifar10 non-IID and (b) SSFL and DisPFL On Cifar100, with increasing sparsity levels

**Quality of the discovered mask in the non-IID setting**  In this section we ask the question, what is the effect of choosing a global common mask on the performance of the local model $f_k$ on it's local data distribution $\mathcal{D}_k$ in a FL setting with highly uneven proportion of data at different sites. We plot the performance of the trained local sparse models with SSFL in Figure 3 in Appendix-A.2 on the left and the proportion of data available at each sites on the right. We note that, even with large discrepancy in the proportion of available data at these different sites, the global mask learned transfers enough information from the other sites to result in consistently high test accuracy on the local data. Moreover, the masks discovered is also robust to significant imbalances of the class distribution at these local sites as well. We illustrate the accuracy of local sparse models and their corresponding class distribution for 10 random clients and also the top and the bottom 10 performing clients in Fig A.4 in Appendix A.4.

## 5 CONCLUSION

In this paper, we proposed SSFL, a novel federated learning paradigm that collaboratively trains highly sparse models while maintaining robust performance. This framework can be effectively used to reduce the communication cost and improve bandwidth during decentralized federated training. To achieve this, we successfully extend a gradient based parameter importance criterion to the decentralized Federated Learning setting in contrast to earlier attempts that resulted in sub-optimal results, and experimentally verify the efficacy of our method. Our parameter saliency score captures the local data characteristics at client sites resulting in a client data aware global sparse mask. Consequently, we demonstrate highly competitive performance on a range of standard and more complex tasks. Leveraging sparse mask discovery before training, SSFL only requires transmission of sparse parameters between clients resulting in savings in communication time and bandwidth requirements. We also test our method on a real world FL framework where we deploy our sparse decentralized FL training resulting in improvement in wall-time performance, which we report in the appendix.

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

# A APPENDIX

## A.1 DETAILED ANALYSIS OF CONNECTION IMPORTANCE

In this section, we provide a detailed analysis of the gradient based connection saliency metric that we build on in this work. The primary idea in neural network pruning research is that neural networks often have more parameters than necessary, and it's possible to achieve similar performance with a smaller network. This concept, initially suggested by Reed (1993), is believed to enhance generalization, as later supported by Arora et al. (2018). The goal of this process is to develop a more efficient, sparse network without compromising the accuracy of the original, more complex network. Let's start by framing neural network pruning as an optimization problem.

We consider a neural network $f$ at a site $k$, parametrized by $\boldsymbol{w} \in \mathbb{R}^d$ with the parameters $\boldsymbol{w}_0$ at initialization. The objective at site $k$ given a training set $\{(\mathbf{x}_i, y_i)\}_{i=1}^M \sim \mathcal{D}$ the local data distribution can be framed as the following minimization problem of the empirical risk:

$$\mathcal{L} = \frac{1}{M} \sum_i \ell(\boldsymbol{w}_0; \boldsymbol{x}_i, y_i) \tag{8}$$

A sub-network within this network is defined as a sparse version of this network with a mask $\mathbf{m} \in \{0, 1\}^{|\boldsymbol{w}|}$ that results in a masked network $f(\boldsymbol{w} \odot \mathbf{m}; \boldsymbol{x}_i, y_i)$. In general, gradient-based connection importance criterions consider the Taylor expansion of the change in the loss to approximate the importance of a neuron or connections. Varying forms of this have been proposed in the literature, the first, probably three decades ago by Mozer & Smolensky (1988) and more recently by Lee et al. (2018); De Jorge et al. (2020) and reformulated in Frankle et al. (2021); Shen et al. (2022), of which we provide a detailed exposition below in line with the works above: Since the goal is to measure the importance or saliency of each connection an auxiliary indicator variable $\mathbf{c} \in \{0, 1\}^d$ is introduced representing the degree of importance of parameter $\mathbf{w}$. Given a sparsity level $\lambda$, we can rewrite equation-8 as follows:

$$\mathcal{L} = \frac{1}{M} \sum_i \ell(\mathbf{c} \odot \boldsymbol{w}_{k,0}; \boldsymbol{x}_i, y_i) \tag{9}$$

$$\text{s.t.} \quad \boldsymbol{w} \in \mathbb{R}^d, \quad \mathbf{c} \in \{0, 1\}^d, \quad \|c\|_0 \leq \lambda$$

The main goal of the above formulation is to use $c_j$ as a metric for the importance of each connection $w_j$, where $w_j$ indicates the $j$-th connection and $c_j$ the corresponding auxiliary scalar which enables us to determine the saliency of each connection by measuring its effect on the loss function. Therefore, the effect of removing the connection $j$ is captured by:

$$\Delta\mathcal{L}_j(\boldsymbol{w}; \mathcal{D}) = \mathcal{L}(1 \odot \boldsymbol{w}; \mathcal{D}) - \mathcal{L}((\mathbf{1} - \boldsymbol{e}_j) \odot \boldsymbol{w}; \mathcal{D}) \tag{10}$$

where $\boldsymbol{e}_j$ is the indicator vector for element $j$ and $\mathbf{1}$ is the vector of ones of dimension $d$. It can be noted that at this stage, computing the $\mathcal{L}_j$ for each $j \in \{1, ..., d\}$ is computationally restrictive as it requires $d + 1$ forward passes over the data, which would be in the order of millions. Moreover, since $\mathbf{c}$ is binary, $\mathcal{L}$ is not differentiable with respect to $\mathbf{c}$. As $\mathbf{c}$ is an auxiliary variable trying to measure the importance of $\boldsymbol{w}$ on the loss in the discrete setting, the binary constraint on $\mathbf{c}$ can be relaxed and $\mathcal{L}_j$ can be estimated by the derivative $g_j(\boldsymbol{w}; \mathcal{D})$ of $\mathcal{L}$ with respect to $c_j$. Consequently, the effect of connection $j$ on the loss is then:

$$\Delta\mathcal{L}_j(\boldsymbol{w}; \mathcal{D}) \approx g_j(\boldsymbol{w}; \mathcal{D}) = \frac{\partial L(\mathbf{c} \odot \boldsymbol{w}; \mathcal{D})}{\partial c_j}\bigg|_{\mathbf{c}=1} \tag{11}$$

$$= \lim_{\delta \to 0} \frac{\mathcal{L}(\mathbf{c} \odot \boldsymbol{w}; \mathcal{D}) - \mathcal{L}((\mathbf{c} - \delta\boldsymbol{e}_j) \odot \boldsymbol{w}; \mathcal{D}}{\delta}\bigg|_{\mathbf{c}=1}$$

It should be noted that $\frac{\partial\mathcal{L}}{\partial c_j}$ represents the infinitesimal counterpart of $\Delta\mathcal{L}_j$. This denotes the rate at which the loss function $\mathcal{L}$ changes in response to a minute alteration in $c_j$, shifting from 1 to $1 - \delta$. This concept essentially gauges the impact on the loss function resulting from a slight adjustment in the weight $w_j$ by the factor $\delta$. It is also crucial to distinguish between $\frac{\partial\mathcal{L}}{\partial c_j}$ and $\frac{\partial\mathcal{L}}{\partial w_j}$; the latter signifies the gradient with respect to the weight $w_j$, which is a different measure.

**Reformulation of the Importance Criterion:** The core idea of the importance criteria presented in this section is to preserve the parameters that has maximum impact on the loss under perturbation. The importance criterion in equation-11 can be reformulated keeping in mind the above analysis in a more straight-forward way employing the chain rule:

$$g_j(\boldsymbol{w}; \mathcal{D}) = \frac{\partial \mathcal{L}(\boldsymbol{w} \odot \mathbf{c})}{\partial \mathbf{c}}\bigg|_{\mathbf{c}=1} = \frac{\partial \mathcal{L}(\boldsymbol{u})}{\partial \boldsymbol{u}} \cdot \frac{\partial \boldsymbol{u}}{\partial \mathbf{c}} = \frac{\partial \mathcal{L}(\boldsymbol{w})}{\partial \boldsymbol{w}} \odot \boldsymbol{w} \tag{12}$$

Where we use a substitution of variable $\boldsymbol{u} = \boldsymbol{w} \odot \mathbf{c}$ which implies, $\partial \boldsymbol{u}/\partial \mathbf{c} = \boldsymbol{w}$ and $\frac{\partial \mathcal{L}(\boldsymbol{u})}{\partial \boldsymbol{u}} = \frac{\partial \mathcal{L}(\boldsymbol{w})}{\partial \boldsymbol{w}}$ at $\mathbf{c} = 1$. Our primary goal is to identify significant connections in the model at the beginning of training, in line with studies like Lee et al. (2018); De Jorge et al. (2020); Shen et al. (2022); Frankle et al. (2021). To achieve this, we utilize the magnitude of the gradient $|g_j(\boldsymbol{w}; \mathcal{D})|$ from equation-11 as an indicator of connection importance. A higher value of this magnitude suggests that the corresponding connection $w_j$, associated with $c_j$, has a substantial influence on the loss function, implying its significance in the training process and should be preserved. Consequently, a saliency score for each connection can be calculated based on this principle and normalized, although normalization is not essential as we are interested in the ordering only, as follows:

$$s_j = \frac{|g_j(\boldsymbol{w}; \mathcal{D})|}{\sum_{i=1}^{d} |g_i(\boldsymbol{w}; \mathcal{D})|} \tag{13}$$

From this score, the top-$\lambda$ most important connections are preserved, where $\lambda$ is the number of parameters to be preserved.

## A.2 EXPERIMENTAL SETTING AND FURTHER RESULTS

The performance of our proposed method is assessed on three image classification datasets: CIFAR-10, CIFAR-100 Krizhevsky et al. (2009), and Tiny-Imagenet. We examine two distinct scenarios to simulate non-identical data distributions among federating clients. Following the works of Hsu et al. (2019), we use Dir Partition, where the training data is divided according to a Dirichlet distribution $Dir(\alpha)$ for each client, and the corresponding test data for each client is generated following the same distribution. We take an $\alpha$ value of 0.3 for CIFAR-10, and 0.2 for both CIFAR-100 and Tiny-Imagenet. Additionally, we conduct an evaluation using a pathological partition setup, as described by Zhang et al. (2020), where each client is randomly allocated limited classes from the total number of classes. Specifically, each client holds 2 classes for CIFAR-10, 10 classes for CIFAR-100, and 20 classes for Tiny-Imagenet in our setup similar to Dai et al. (2022).

**Implementation of Baselines:** We describe most of the implementation detail for the baselines in section 4. We use the official implementaion of DisPFL by Dai et al. (2022) available in their github use similar hyper-parameters for both DisPFL and SSFL. FedPM trains sparse random masks instead of training model weights and finds an optimal sparsity by itself Isik et al. (2022). We use the official implementation of FedPM available in their github page for the paper. For fair comparison, we use the cifar10 non-IID split that we use in this work with the sample $\alpha$ values as described above in this section. For top-k and random-k baselines, we follow a training strategy that is similar to SSFL and in essense to FedAvg McMahan & Ramage (2017) or DisPFL Dai et al. (2022) in terms of the weight aggregation mechanism. For the top-$k$ implementation, the top-$k$ operator is applied to the trained weights to be transmitted and shared between the clients after $T$ steps of local training and only the top-$k$ weights are shared and aggregated. To establish random-$k$ as a random sparse training, similar to our proposed sparse FL and DisPFL training method we generate a random-$k$ mask at each local site and train and aggregate the weights corresponding to that mask after every $T$ steps of training. We note that unlike SSFL, DisPFL or random masking, at the end of training top-$k$ does not result in sparse local models with the benefits of sparse models such as faster inference.

**Convergence under longer training and more communication rounds** We report the convergence plots for SSFL and other methods in Figure-4 when trained for a total of $R = 500$ communication rounds. As Ditto does not properly converge within 500 rounds, we conduct further experiments for SSFL, DisPFL and Ditto on a much longer $R = 800$ communication rounds to analyze the convergence of these methods. We demonstrate the convergence plots in Figure-5.

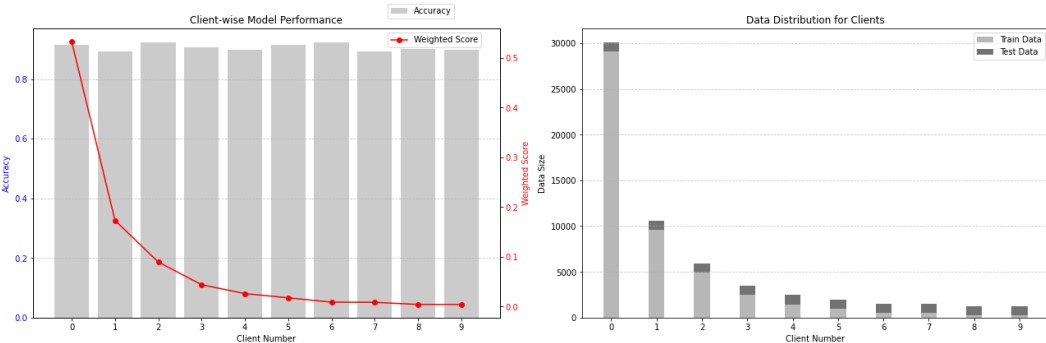

Figure 3: Accuracy of the local client models on the left and the proportion of the total data available at the corresponding sites on the right. The discovered global mask enables training of performant models even on client sites with low data.

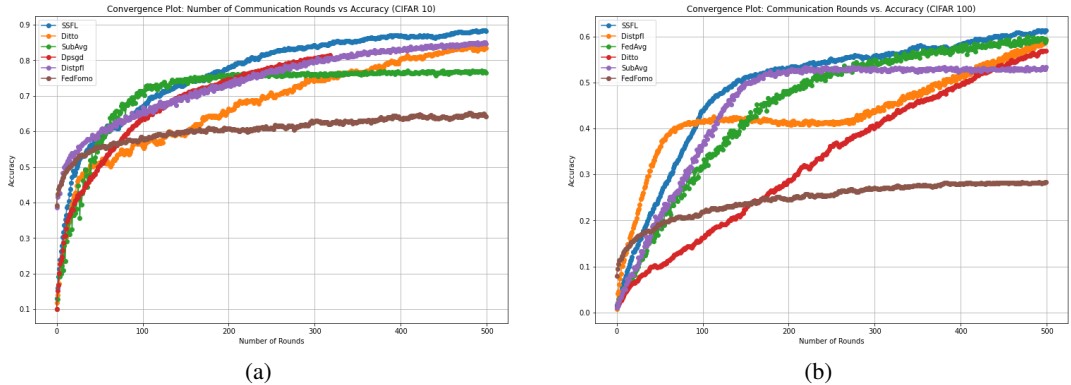

Figure 4: Comparison of convergence rates of SSFL with similar decentralized federated and sparse federated learning methods(a) On CIFAR10 (b) on CIFAR100 for total $R = 500$ communication rounds.

**What happens if we find unique local masks instead of a global mask?** In this work we leverage the saliency criterion in equation-4 and extend it to the non-IID Federated Learning setting in equation-6 to generate common global masks for models and train the models in the non-IID federated learning setting. An interesting question to ask is, what happens when unique local masks are devised from the saliency criterion and the local models are free to train their own models based on this mask? We briefly explored local masking and training at a sparsity level of $50\%$ where the trained model demonstrated an accuracy of $70.22\%$. Although this is competitive with some methods such as Isik et al. (2022) in terms of performance, but suboptimal when compared to global masking. We hypothesize that this might be due to the complex way in how completely non-overlapping masks and weights interacted and updated with each other after aggregation along with random sampling of clients, which is sidestepped when training client models with the same mask. It might be the case that the local models would drift far away in the restricted (sparse) solution space and their aggregation resulting in subpar models. This would indeed be a an interesting avenue for future research and requires extensive experimentation in this direction of exploration to arrive at a conclusive answer.

A.3 DATA AND CLASS DISTRIBUTION AT CLIENT SITES

In this section we plot some samples of the uneven class distribution at different client sites for our experiments on the CIFAR10 and the CIFAR100 dataset under the Dirichlet distribution of the class labels at different sites in Fig 6 .

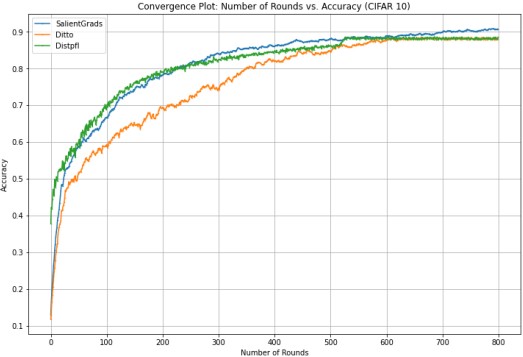

Figure 5: Comparison of convergence rates of SSFL with similar federated (Ditto) and sparse federated learning (DisPFL) methods(a) On CIFAR10 (b) on CIFAR100 for a longer $R = 800$ communication rounds.

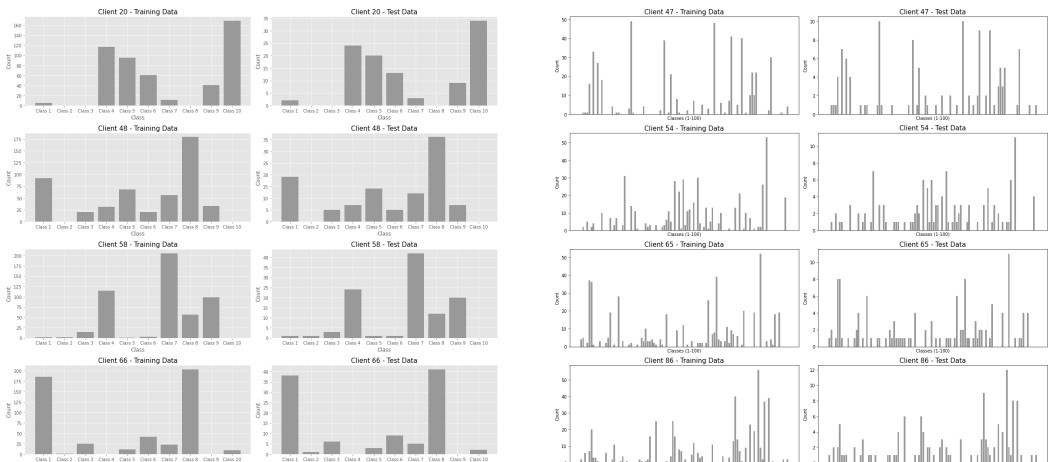

Figure 6: Class distribution from sampled clients under Dirichlet Distribution. (a) For the CIFAR10 data with $\alpha = 0.3$ (b) For the CIFAR100 data with $\alpha = 0.2$.

### A.4  LOCAL CLIENT ACCURACY AND CLASS DISTRIBUTION

We plot the class distribution with $\mathrm{Dir}(\alpha)$ on the CIFAR10 dataset with $\alpha = 0.3$ for 10 random clients and their final accuracy in Fig 7 (a). In Fig (b) we plot the top-10 clients in terms of their final test accuracy and in Fig-(c) the bottom 10 clients in terms of final test accuracy. We notice that, SSFL finds masks that result in consistent local model performance and even in the bottom 10 clients, the performance remains respectable.

### A.5  EXPERIMENTS ON THE REAL WORLD COINSTAC SYSTEM

We performed the experiments using Amazon Web Services (AWS), by creating multiple instances to perform the federated training. We used a general framework called COINSTAC Plis et al. (2016), which is a open-source federated learning solution that focuses on analysis of imaging data, with an emphasis on facilitating collaboration between research institutions throughout the world. To simulate a real world FL training scenario, we selected the AWS nodes from 5 different locations throughout the world: North Virginia, Ohio, Oregan, London, and Frankfurt. We performed experiment on these five different sites, leaving additional experiments for a more exhaustive future work due to the limitation in budget and time.

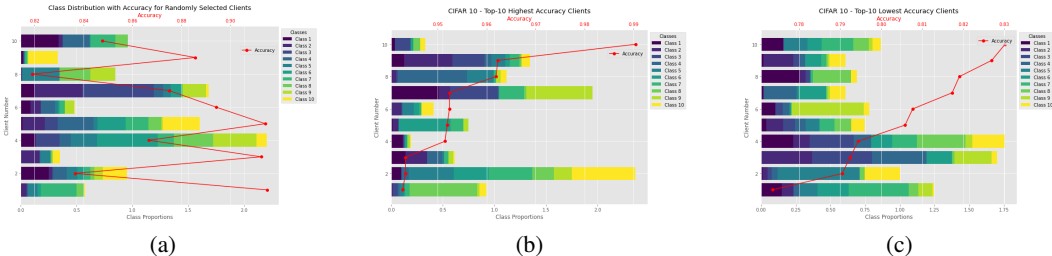

(a)                                        (b)                                        (c)

Figure 7: (a) Class distribution with Dir(0.3) for the CIFAR10 dataset for 10 random clients and their final local test accuracy (b) the top-10 clients in terms of their final test accuracy (c) the bottom-10 clients in terms of their final test accuracy

**Experimental Details**  In our experiments, we used the CIFAR-10 dataset. For the model architecture, we train a range of ResNet models with increasing size and depth (ResNet20, ResNet32, ResNet44, ResNet56, ResNet110 and ResNet1202) to evaluate the performance of the algorithm on different scale of model parameters.

In these experiments we compared SSFL to the standard `FedAvg` with no compression and demonstrate the viability of the method in terms of computational efficiency and performance stability in a real-world FL framework.

**Evaluation in standard federated learning scenarios**  We first present the performance of the proposed approach on the CIFAR-10 dataset, which is evaluated in a distributed setting with 5 different local client models with varying size or depth of ResNet models. Fig 8 shows the communication-time in seconds for SSFL and *FedAvg* models for different ResNet architectures in a logarithmic plot. We report the mean cumulative communication time (i.e, the time taken by the server model to gather all the weights for each mini-batch). The sparsity for all experiments was fixed to be around 90%. The average communication time between the two techniques, the total number of model parameters and the corresponding speed ups in wall-clock time are demonstrated in Table 4. Moreover, another important metric when building sparse models is the model performance. We also highlight the performance of our technique in terms of accuracy and similar metrics for different model architectures using CIFAR-10.

| Architecture | Number of Parameters | Accuracy | Communication Time (s) | | Speed up |
|---|---|---|---|---|---|
| | | | FedAvg | SSFL | |
| ResNet20 | 0.27M | 84.62% | $0.188 \pm 0.04$ | $0.147 \pm 0.04$ | 1.27 |
| ResNet32 | 0.46M | 90.52% | $0.285 \pm 0.04$ | $0.238 \pm 0.02$ | 1.20 |
| ResNet44 | 0.66M | 89.65% | $0.409 \pm 0.06$ | $0.328 \pm 0.04$ | 1.24 |
| ResNet56 | 0.85M | 93.74% | $0.531 \pm 0.07$ | $0.407 \pm 0.06$ | 1.30 |
| ResNet110 | 1.7M | 93.25% | $1.812 \pm 0.33$ | $0.781 \pm 0.13$ | 2.32 |

Table 4: Performance comparison between *FedAvg* and SSFL on different ResNet architectures based on communication time.

We observe in Fig 8 that SSFL outperforms *FedAvg* for every scale of ResNet models and the benefits of sparsity grows with model size. It was observed that larger models tends to drastically benefit from our technique with almost $2.5\times$ improvement in the communication time. In terms of accuracy, we observe with 90% sparsity, we get a stable performance for different set of ResNet models. This is especially significant due to the real world nature of the COINSTAC framework with constrained resources and computation overheads Plis et al. (2016).

A.6    EVALUATION OF SSFL ON REAL WORLD NEUROIMAGING DATASET

In this section we evaluate SSFL on a real-world neuroimaging dataset using the framework explained in A.5. We report the results on Fig:9

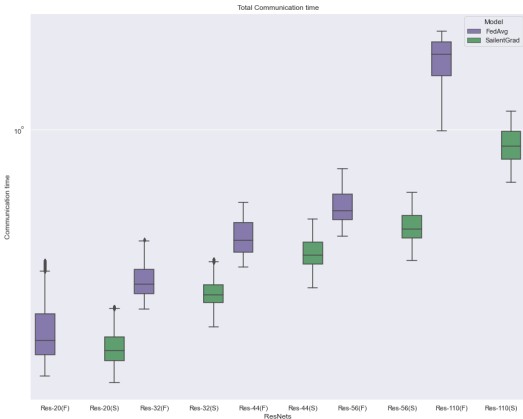

Figure 8: Performance comparision of *FedAvg* vs SSFL on different ResNet architectures based on communication time.

**Explanation of the ABCD dataset and its role in NeuroImaging**   The current study utilizes the dataset from version 2.01 of the Adolescent Brain Cognitive Development (ABCD) study, available at [ABCD Study Website](https://abcdstudy.org/). The ABCD dataset encompasses data from over 11,800 children aged between 9 and 11 years from 21 sites across the US, including multiple MRI scans from two imaging sessions (baseline and second-year follow-up). Additional details on the distribution of data across each of the 21 sites can be referenced from Fig9(b). This extensive dataset covers a diverse array of demographic and health backgrounds. For each participant in the ABCD study, full written informed consent from the parents and assent from the child was duly obtained, adhering to protocols approved by the Institutional Review Board (IRB). The National Institute of Mental Health Data Archive (NDA), accessible at, shares the ABCD dataset. The NDA provides open-source datasets from a broad spectrum of research projects across various scientific domains, promoting collaborative science and discovery.

The structural MRI (sMRI) data underwent segmentation, transforming into probability maps distinguishing gray matter, white matter, and cerebrospinal fluid using Statistical Parametric Mapping 12 (SPM12) software. Subsequently, the gray matter images were standardized, modulated, and smoothed through Gaussian kernel convolution with a Full Width at Half Maximum (FWHM) of 10 mm. The resulting preprocessed gray matter volume images were represented in a voxel space measuring 121×145×121, with each voxel having dimensions of 1.5×1.5×1.5mm³. Fig11 demonstrates the samples of the processed MRI scans that are used as the input to the model.

**Architecture, Hyperparameters and experimental details**   In this section, we provide a comprehensive view of the architecture, hyperparameters and the experimental setup used for evaluating the SSFL method on a real-world neuroimaging ABCD dataset. Our study focuses on the task of classifying the gender based on MRI scans, by employing a 3D variant of the well-known AlexNet model Krizhevsky et al. (2012). The 3D variant was referenced from Abrol et al. (2021), which has a specific channel configuration for the convolutional layers set as: 64C-128C-192C-192C-128C, where 'C' denotes channels. This model was particularly found to be effective for gender classification on similar neuroimaging datasets of MRI scans.

We optimized the learning rate for this task through an exhaustive search ranging from LR=0.001 to 0.000001, achieving a delicate balance between rapid convergence and fine-tuning during training. We employed a batch size of 32 and a learning rate decay factor of 0.998 was applied. W applied varying sparsity levels, ranging from 50%, 60%, 70%, 80%, and 90% to assess the overall performance. A split of 80/20 was used for training and testing for each individual site. Our training consisted of 5 epochs with 200 communication rounds. We compared our SSFL method with the standard DistPFL and FedAvg-FT model, and the results are demonstrated in Fig 9(a), where we observe improved performance over DisPFL. Moreover, in Fig 10 we notice that the performance of local models trained with the SSFL the framework remains mostly robust across non-IID state of the data local data and performs consistently well.

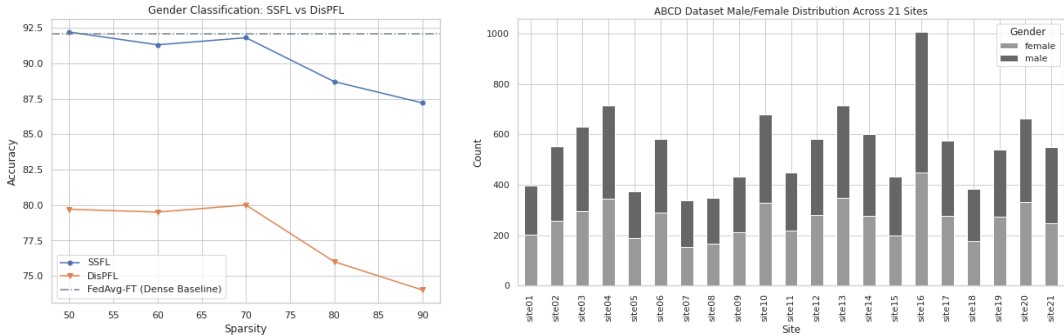

Figure 9: (a) Comparison of methods for gender classification using MRI Scans of ABCD dataset. (b) Distribution of gender classes of the non-IID CIFAR100 dataset across clients.

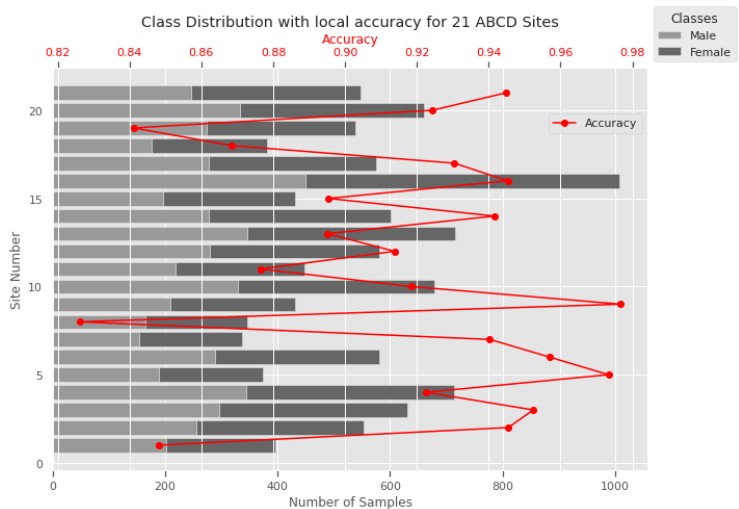

Figure 10: Gender differences in each of the 21 ABCD sites along with the performance of the model.

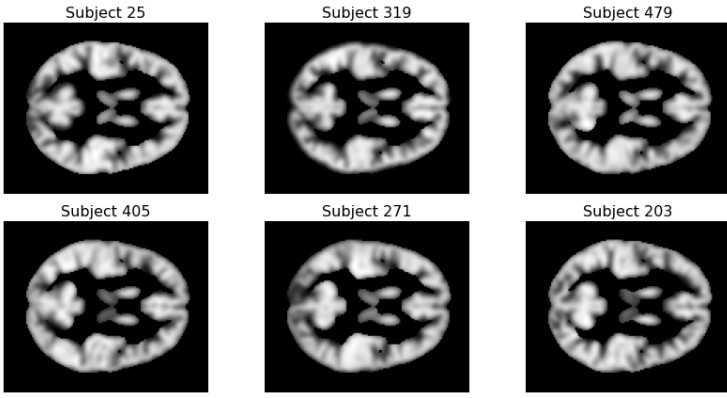

Figure 11: Sample structural MRI brain image of brains used in gender classification.

