# OpenReview forum: "From Random to Relevant: Harnessing Salient Masks in Non-IID Federated Learning"
_ICLR.cc/2024/Conference — Submitted to ICLR 2024_

### Official Review · Reviewer_p421 · 2023-10-31

**Soundness:** 2 fair
**Presentation:** 2 fair
**Contribution:** 1 poor
**Rating:** 3
**Confidence:** 3

**Summary:**

This paper explores how to settle limited computation and communication resources in federated learning. It thereby proposes SSFL, an approach to identify a subnetwork before the model training. Specifically, the parameters in the subnetwork are the largest $k$ values of saliency scores calculated by the absolute element-wise product of the weights and the gradients. The experiments validate the proposed SSFL can reduce the communication overhead as well as improve the model performance when compared to the existing works.

**Strengths:**

1. This paper introduces a simple and efficient approach to subnetwork extraction.
2. This paper comprehensively reviews the relative works.

**Weaknesses:**

1. The contribution of this paper is trivial. Although the proposed method seems efficient because the mask is found in the beginning, I don't think the proposed approach makes sense. Before the model training, the parameters $w_0$ are generated at random. Denote the gradient by $g_0$ for the initial model. Intuitively, when the model converges, the important parameters are with relatively large values, while the ignorable ones are close to 0. As the mask is merely generated in the beginning, I cannot see why Eq. (3) can find a reasonable mask. In other words, I cannot see the differences when the mask is generated arbitrarily.
2. In addition to the mask initialization, the rest of the design is consistent with FedAvg. Following the first point, I cannot see the significance of this work.
3. The authors mention the work based on decentralized FL. After reading the paper, I don't have an idea why it can work under peer-to-peer network architecture. Instead, it is solely workable with the client-server settings.
4. The experiments only show the result on a sparsity level of 50\%, which is not convincing. I would like to see the performance results when the sparsity level is at 10\% or smaller. According to Figure 4, I notice DisPFL and Ditto are yet to converge. I wonder about their final results when they converge.

**Questions:**

Please address my concerns listed in the weaknesses. In addition, I suggest the authors conduct an empirical study to show the performance differences when the mask is randomly drawn in the beginning.

---

> ### Author Response · Authors · 2023-11-21
> **Response to the reviewer p421 (part 1/3)**
>
> We thank the reviewer for his review and helpful suggestions. We appreciate the recognition of our approach's efficiency and the thoroughness of our literature review by the reviewer. Regarding the validity of our masking technique, we believe there might have been a miscommunication in our initial manuscript. The criterion for our masking process is firmly established in theory and on a comprehensive body of literature.
>
> To address this gap, we have expanded our explanation and provided additional evidence to substantiate the robustness and relevance of our method in the revised manuscript (Section 3.1 and Appendix A.1). This should clarify any uncertainties and demonstrate the validity of our approach clearly. We also address all the points brought up by the reviewer and make necessary changes in the updated manuscript (in purple).
>
> ### 1. Regarding the efficacy of the Masking criterion
> > 1. "the proposed method seems efficient because the mask is found in the beginning, I don't think the proposed approach makes sense. Before the model training, the parameters are generated at random. .... In other words, I cannot see the differences when the mask is generated arbitrarily."
>
> We acknowledge that our initial reliance on references for presenting the saliency criterion might have led to the reviewer's conclusions. To address this, we have thoroughly revised Section 3.1 of our manuscript, offering a comprehensive explanation of the saliency criterion, which is well-supported by extensive literature [1,2,3,4,5]. Furthermore, we present a detailed exposition of the process in Appendix A1.
>
> These revisions aim to clarify the mask generation process, explaining its methodical nature and demonstrating its effectiveness over random masking. Additionally, to substantiate our claims, we have included experiments comparing our method with random masking in the latter part of this response. Below we briefly provide an explanation of the masking process based on [1,2,3,4,5] here:
>
> ### Explanation of the validity of Connection Importance Criterion and Mask Generation
> Since the goal is to measure the importance or saliency of each connection an auxiliary indicator variable $\mathbf{c} \in \\{ 0,1 \\}^d$ is introduced representing the degree of importance of parameter $\mathbf{w}$. Given a sparsity level $\lambda$, the objective can be re-written as follows:
>
> \begin{align}
>     \mathcal{L} &= \frac{1}{M} \sum_i \ell ( \mathbf{c} \odot \mathbf{w}; \mathbf{x}_i, y_i) \quad \text{s.t.} \quad \mathbf{w} \in \mathbb{R}^d, \quad \mathbf{c} \in \\{ 0,1 \\}^d, \quad ||c||_0 \leq \lambda \notag
> \end{align}
>
> The main goal of the above formulation is to use the $j$-th element $c_j$ of $\mathbf{c}$ as a metric for the importance of each connection $w_j$, where $w_j$ indicates the $j$-th connection, which enables us to determine the saliency of each connection by measuring its effect on the loss function.
>
> The core idea of the importance criteria is to preserve the parameters that have maximum impact on the loss under perturbation. The importance criterion is then formulated as:
> $$
>     \mathbf{g}_j(\mathbf{w}; \mathcal{D})
>     = \frac{\partial \mathcal{L} (\mathbf{w} \odot \mathbf{c})}{\partial \mathbf{c}}  \\
>     = \frac{\partial \mathcal{L} (\mathbf{u})}{\partial \mathbf{u}} \cdot \frac{\partial \mathbf{u}}{\partial \mathbf{c}} \\
>     = \frac{\partial \mathcal{L} (\mathbf{w})}{\partial \mathbf{w}} \odot \mathbf{w}
> $$
> Where we evaluate the above at $\mathbf{c}=1$ and use a substitution of variable $\mathbf{u} = \mathbf{w} \odot \mathbf{c}$ which implies, $\partial \mathbf{u}/ \partial \mathbf{c} = \mathbf{w}$ and $\frac{\partial \mathcal{L} (\mathbf{u})}{\partial \mathbf{u}} = \frac{\partial \mathcal{L} (\mathbf{w})}{\partial \mathbf{w}}$ at $\mathbf{c = 1}$ and relax the binary constraint on the indicator variables $\mathbf{c}$ to take on real values $\mathbf{c} \in [0,1]^d$. To achieve our primary objective of identifying significant connections towards the start of training, we utilize the magnitude of the gradient $|\mathbf{g}_j(\mathbf{w}; \mathcal{D})|$ as an indicator of connection importance (note that this is the gradient with respect to $\mathbf{c}$ and not $\mathbf{w}$). A higher value of this magnitude suggests that the corresponding connection $w_j$ associated with $c_j$ has a substantial influence on the loss function, implying its significance in the training process and that the connection should be preserved. Consequently, a saliency score for each connection can be calculated based on this principle as follows:
>
> \begin{align}
>     s_j = \big| g_j(\mathbf{w}; \mathcal{D}) \big| = \bigg| \frac{\partial \mathcal{L} (\mathbf{w})}{\partial \mathbf{w}} \odot \mathbf{w} \bigg|
> \end{align}
>
> We next extend this saliency criterion in section (3.1) in paper based on the data distribution at each site and generate a mask choosing the top-$\lambda$ ranked connections as: $\mathbf{m} = \mathbb{1}[s(w_j) \geq s(w_\lambda)]$

---

> ### Author Response · Authors · 2023-11-21
> **Response to the reviewer p421 (part 2/3)**
>
> ### 2. Significance of this work
> > 2. In addition to the mask initialization, the rest of the design is consistent with FedAvg. Following the first point, I cannot see the significance of this work.
> >
> In light of the above explanation on the validity of our mask generation process and its grounding in extensive literature we note the significance of our work here.
> - From our reading of the literature, the current leading works in terms of sparsity vs accuracy trade-offs, and hence communication efficiency in the realistic non-IID setting is DisPFL and FedPM [6, 7]. As FedPM trains a mask on the initial weights and never trains the weights, it loses accuracy in the process, similar to the works with supermask [10].
> - DisPFL on the other hand finds a *random mask* at initialization and periodically updating the mask trains the model in a manner similar to FedAvg as well. One of the major goals of our work is to demonstrate that, in contrast to DisPFL's random initial masking and updating, finding a salient mask at initialization results in a **significantly** better sparsity vs accuracy trade-offs in the non-IID setting.
> - We also note that previous work [8,9] reported unsuccessful uses of the saliency criterion in the FL setting, whereas our proposed extensions for the saliency criterion in this work to the non-IID setting effectively helps the method to outperform competing similar methods.
>
> ### 3. SSFL in the Decentralized setting:
> > 3. The authors mention the work based on decentralized FL. After reading the paper, I don't have an idea why it can work under peer-to-peer network architecture. Instead, it is solely workable with the client-server settings.
>
> We appreciate the reviewer's interest in the applicability of our method in a decentralized setting. Since the reviewer does not mention *why this method would not work in a decentralized setting*, we provide a general explanation. In this work we do not assume the existence of a server, however, our proposed technique should be conceptually viable for both a server-client setup and a decentralized setup. We only require most client participation during initial weights alignment and transmitting and receiving saliency scores and sample count. These operations are required only once, at the start of training, for alignment purposes and would be pulled/gathered from all the clients at each client $k$. We make changes to clarify the exposition in Section 3.2.
>
> Moreover, focusing on line-12 of Algorithm-1, we note that each client gathers the current masked weights from a subset of all other clients and trains the model. This is true for all the $K'$ selected clients, where they all gather weights among themselves.
>
> The work however, does not focus on the engineering details of deploying this in a real-world decentralized setting or go into exhaustive details as that is beyond the scope of our work and are deployment details. However, taking into reviewer's concern we have added modifications in the algorithm and updated the methods section 3.2 to help clarify. We are however happy to discuss if the reviewer has any specific concerns regarding SSFL's viability in the decentralized setting.
>
> ------
>
> #### 4. Further Clarification on Initially Provided Performance Metrics Across Different Sparsity Levels
> > 4. The experiments only show the result on a sparsity level of 50%, which is not convincing. I would like to see the performance results when the sparsity level is at 10% or smaller.
>
> We would like to point out that we have reported the performance of SSFL on varying sparsity levels of 50, 70, 80, 90 and 95 percent in Figure 2 of the initial paper submission and compared with DisPFL at the same sparsity levels as well as with the dense baseline FedAvg-FT. Thus, the results do include $<10\%$ density levels as asked by the reviewer.
> However, following reviewer's sugestion, we now also include results from two additional methods: SubFedAvg and FedPM. Please note that FedPM, a more recent method, automatically chooses the most optimal sparsity and does not allow sparsity sweeps. We present the results below:
>
> | Methods               | Sps 0% | 50% | 70% | 80% | 90% | 95% |
> |----------|----------|----------|----------|----------|----------|----------|
> | **FedAvg-FT** (dense) | 88.02 | | | | | |
> | **SSFL**              | | **88.29** | **87.57** | **87.43** | **84.58** | **84.18** |
> | **DisPFL**            | | 85.82 | 83.39 | 82.93 | 76.17 | 75.00 |
> | **SubFedAvg**         | | 0.7970 | 0.8489  | 0.7818 | 0.7813 | 76.77 |
> | **FedPM**             | (optimal sps) 57.30 | N/A | N/A | N/A | N/A | N/A |
>
> **Table:** *Accuracy at given sparsity levels. We note that FedPM optimally chooses its sparsity level and does not allow sparsity sweeps hence other sparsity levels are not available N/A.*

---

> ### Author Response · Authors · 2023-11-21
> **Response to the reviewer p421 (part 3/3)**
>
> --------
> ### Longer training and convergence properties
> > According to Figure 4, I notice DisPFL and Ditto are yet to converge. I wonder about their final results when they converge.
>
> In response to the reviewer's suggestion, we conduct extended duration training with $R=800$ communication rounds for Ditto, DisPFL, and SSFL, ensuring convergence, and present the plot in Figure 5 of Appendix A.2 in the revised manuscript.We observe slight improvement in performance for all three methods including our method SSFL (where the improvement was more noticeable), however the ordering remains same, i.e. SSFL performing better than DisPFL and Ditto. We finally emphasize that Ditto is presented as a dense baseline for comparison and is not a sparse or efficient method.
>
>
> ### SSFL masking compared with random masking
> We appreciate the reviewer's suggestion about performing an empirical study comparing with random masking and acknowledge that this is indeed an interesting study to conduct. Taking into the reviewer's comment we report the comparison with random masking below on the Cifar10 non-IID split data:
>
> | Methods              | 50% | 70% | 80% | 90% | 95% |
> |----------|----------|----------|----------|----------|----------|
> | **SSFL** | **88.29** | **87.57** | **87.43** | **84.58** | 84.18 |
> | **Random** | 41.61 | 39.80 | 19.70 | 10.60 | - |
>
> Since the performance of random masking falls around $10$% at $90$% sparsity, which is the accuracy of a random model on Cifar10, we do not conduct further experiments on higher sparsities during the discussion phase.
>
> ### Final Note to the reviewer:
>
> We have addressed the concerns regarding the saliency criterion and masking mechanism used in our study. The foundation of these approaches is well-supported by theory and corroborated by existing literature. We have included a comprehensive explanation within the paper to clarify any doubts.
>
> Should the revisions and our explanations meet the reviewer's standards, we hope that our responses might change the reviewer's perception of critical points raised initially. We believe following our clarification, the updated paper deserves a reevaluation, and we are open to further discussion should there be any residual concerns.
>
> --------
> ### References:
> [1] Skeletonization: A Technique for Trimming the Fat from a Network via Relevance Assessment. [NIPS 1988]
>
> [2] SNIP: Single-shot Network Pruning based on Connection Sensitivity. [ICLR 2019]
>
> [3] Progressive Skeletonization: Trimming more fat from a network at initialization. [ICLR 2021]
>
> [4] Pruning Neural Networks at Initialization: Why are We Missing the Mark? [ICLR 2021]
>
> [5] When to Prune? A Policy towards Early Structural Pruning [CVPR 2022]
>
> [6] DisPFL: Towards Communication-Efficient Personalized Federated Learning via Decentralized Sparse Training. [ICML 2022]
>
> [7] Sparse Random Networks for Communication-Efficient Federated Learning. [ICLR 2023]
>
> [8] Fedtiny: Pruned federated learning towards specialized tiny models. [arXiv 2022]
>
> [9] Model pruning enables efficient federated learning on edge devices. *IEEE Transactions on Neural Networks and Learning Systems*, 2022.
>
> [10] Deconstructing Lottery Tickets: Zeros, Signs, and the Supermask. [NeurIPS 2019]

---

### Official Review · Reviewer_1qhQ · 2023-10-31

**Soundness:** 3 good
**Presentation:** 3 good
**Contribution:** 2 fair
**Rating:** 8
**Confidence:** 2

**Summary:**

In this work, the authors suggest a novel federated learning paradigm that trains a sparse model while maintaining robust performance. This paradigm can be used to improve bandwidth during decentralized federated training and reduce communication costs. The authors demonstrate the effectiveness of their approach by testing it on multiple datasets (TinyImageNet, Cifar 10, Cifar 100) on various models. The suggested approach surpasses other benchmarks in most of the experiments.

**Strengths:**

I really liked the presentation of this paper, I am not familiar with federated learning but it was easy for me to understand the federated learning and the suggested paradigm. Also, the suggested approach is novel.

**Weaknesses:**

As I mentioned before, this novel approach raises two concerns:

1) While the approach is indeed innovative, it appears to be overly simplistic.
2) The experimental results are promising, but there is potential for improvement, particularly when tested on alternative models and datasets.

**Questions:**

1) Can this approach be applied to Large Language Models (LLMs)? If so, incorporating empirical results would significantly enhance the paper.
2) Is your proposed method connected to the concept of core sets?
----------------------------------
The authors clearly addressed my concerns. Hence, I am raising my score to Accept.

---

> ### Author Response · Authors · 2023-11-21
> **Response to the review of 1qhQ (part 1/2)**
>
> We are grateful to the reviewer for their careful review and kind words regarding the presentation of our paper. We are also grateful for the acknowledgment of the novelty of our approach. We attempt to address the points and questions brought up by the reviewer below:
>
> > While the approach is indeed innovative, it appears to be overly simplistic.
>
> 1. While we respect the viewpoint that our approach may appear overly simplistic, we would like to pose an alternative perspective to consider the elegance of simplicity whilst maintaining effectiveness. The approach introduced by us is indeed intuitive, but it robustly builds upon the existing body of work in this field, extending the sparsity vs accuracy trade-off in the non-IID Federated Learning training regime. We would like to pinpoint that previous contributions tackling comparable problems [1] also hinge on the simplicity of initiating from a random mask and allow clients to update the mask using only the gradient as the update signal. Thus, we would not like to view simplicity as a quality that undermines the approach's value; instead, we view as an inherent strength of the design.
> 2. Furthermore, we would like to emphasize that our mask generation process, while designed in elegant simplicity, does contain nuanced elements in its execution. To provide a deeper understanding, we have expanded upon the process in section 3.1 of our manuscript, spotlighting these fine details. By laying bare the intricate facets of our seemingly simple approach, we hope to convince the reviewer of the depth of our work beneath its surface.
>
> > The experimental results are promising, but there is potential for improvement, particularly when tested on alternative models and datasets.
>
> We thank the reviewer for this suggestion.  To ensure a comprehensive
> evaluation of our method in the initial submission, we have already tested on three distinct datasets, including the larger Tiny-Imagenet and have utilised modern Resnet models, distinctively different from what had been used in similar studies like FedPM [2]. This rigorous approach ensures the broad applicability of our method.
>
> Responding to the invaluable feedback, we have taken one step further. We conducted additional experiments and have enriched our paper with new empirical results comparing our method with other established methods. These further reinforce the robust performance and major improvements our method exhibits across diverse conditions. Please refer to our detailed response in the opening common response.
>
> We believe the additional experiments improved the part of the work that stands up under various testing conditions and comparison with diverse alternative models. We believe that these enhancements will significantly contribute to the reliability and novelty of our paper.
>
> > Can this approach be applied to Large Language Models (LLMs)? If so, incorporating empirical results would significantly enhance the paper.
>
> We certainly appreciate your suggestion about applying this approach to large language models. The idea is intriguing and could open up several interesting research areas. However, given the complexity involved in this process, it would necessitate extensive experimentation and research beyond the scope of our current work. It's an appealing future research direction, and we have added it to our discussion for potential future work in our revised manuscript. We hope that clarifying this in the paper helps make the potential of this study more clear.
>
> > Is your proposed method connected to the concept of core sets?
>
> We recognize that while the concepts are related, they are not identical as per our interpretation of this work. Core sets were originally developed as a technique to approximate a large set of data points with a smaller yet representative subset, preserving desired criteria.  This concept found its early applications in problems related to k-means and k-medians, where the goal was to represent a large dataset with a smaller, yet representative, subset of data points. In the field of sparsity and pruning, this idea was leveraged for pruning and compression in [3], where CNN model filters were grouped by their activation patterns and represented through their coreset characteristics followed by pruning.
> In contrast, our work can be thought of as reliably finding axial subspaces within the model parameter space that encapsulates a good final solution and we don't believe this closely relates with the concept of core sets from our understanding.
>
> We thank the reviewer for taking the time to review the paper and for providing interesting comments and questions. We hope to have addressed the reviewer's concerns and would be happy to answer further queries if there are any.

---

> ### Author Response · Authors · 2023-11-21
> **Response to the review of 1qhQ (part 2/2)**
>
> #### References
> [1] DisPFL: Towards Communication-Efficient Personalized Federated Learning via Decentralized Sparse Training. [ICML 2022]
>
> [2] Sparse Random Networks for Communication-Efficient Federated Learning. [ICLR 2023]
>
> [3] Coreset-Based Neural Network Compression. [ECCV 2018]

---

> > ### Comment · Reviewer_1qhQ · 2023-11-21
> >
> > Thank you for the detailed response. My concerns were addressed, and I raised my score. Great work.

---

> > > ### Author Response · Authors · 2023-11-21
> > > **Thanks for the review**
> > >
> > > We appreciate your reconsideration and positive evaluation of our work. We are pleased to hear that our responses adequately addressed your concerns. Your constructive feedback was invaluable in refining our paper.

---

### Official Review · Reviewer_mEiz · 2023-11-02

**Soundness:** 3 good
**Presentation:** 3 good
**Contribution:** 2 fair
**Rating:** 6
**Confidence:** 5

**Summary:**

The paper proposes a new communication-efficient federated learning (FL) framework, SSFL, that initially finds a binary sparsity mask by using the non-iid dataset across clients and then trains a sparse network extracted by this mask during the FL stage. This reduces the communication cost in the FL stage as each client trains and communications a sparse model at every round.

**Strengths:**

The proposed method is intuitive, easy to grasp, and the paper is well-written. The empirical results suggests an improvement over the selected baselines.

**Weaknesses:**

- While the proposed method, SSFL, outperforms the selected baselines, overall the performance degrades considerably even for 50% sparsity. For standard gradient sparsification methods such as Top-k[1], Random-k[2], or rTop-k[3], there is no significant performance loss up to 90% sparsity with error feedback. I wonder if the authors have any explanation as to why SSFL and the selected baselines degrade the accuracy even with small sparsity ratios like 50%. Is this because only a fixed sparse portion of the model is trained while the rest is kept at zero? If so, would it be possible to let each client find their own mask rather than aggregating them to have one global mask for all the clients. This way, different portions of the model could be trained by different clients at the cost of losing the communication gain from server to client communication -- which is typically not the main bottleneck compared to client to server communication especially given that the current approach results in considerable accuracy loss.

- I also think comparing SSFL to these alternative sparse methods (Top-k[1], Random-k[2], or rTop-k[3]) would provide a more complete picture to the readers. Also, other methods for finding optimal global sparsity masks should be added as baselines as well such as FedPM [4].

[1] Lin, Yujun, et al. "Deep Gradient Compression: Reducing the Communication Bandwidth for Distributed Training." International Conference on Learning Representations. 2018.

[2] Konečný, Jakub, et al. "Federated learning: Strategies for improving communication efficiency." arXiv preprint arXiv:1610.05492 (2016).

[3] Barnes, Leighton Pate, et al. "rTop-k: A statistical estimation approach to distributed SGD." IEEE Journal on Selected Areas in Information Theory 1.3 (2020): 897-907.

[4] Isik, Berivan, et al. "Sparse Random Networks for Communication-Efficient Federated Learning." The Eleventh International Conference on Learning Representations. 2022.

**Questions:**

Please see Weaknesses.

---

> ### Author Response · Authors · 2023-11-21
> **Response to the review of mEiz (part 1/2)**
>
> We appreciate the reviewer's careful review and valuable suggestions. We are grateful that they found the paper to be intuitive and easy to comprehend, and are pleased that they recognized our performance improvement over the baselines. In response to the reviewer's feedback, we have addressed each of the points and updated the manuscript accordingly. All changes have been highlighted in purple for convenience.
>
> -----
> ### Difference with recommended baselines and new results
>
> > While the proposed method, SSFL, outperforms the selected baselines, overall the performance degrades considerably even for 50% sparsity. For standard gradient sparsification methods such as Top-k[1], Random-k[2], or rTop-k[3], there is no significant performance loss up to 90% sparsity with error feedback. I wonder if the authors have any explanation as to why SSFL and the selected baselines degrade the accuracy even with small sparsity ratios like 50%. Is this because only a fixed sparse portion of the model is trained while the rest is kept at zero?
>
> We first note that the works [1,3] are in the distributed SGD and are not what we consider to be in FL setting, and while the work [3] is conducted in the FL setting, it works under the IID assumption for data distribution. We  summarize the biggest differences between the works mentioned by the reviewers and our work and the baselines below:
>
> 1. **Distributed SGD vs Federated Learning with real world assumptions:**
>     - We would like to point out that some of the studies mentioned by the reviewer, specifically, works [1,3] focus on *gradient sparsification* techniques in a distributed, non-federated setting and are tested under IID assumptions, leading to divergent results. As a result, the sparsity vs accuracy trade-offs in those settings are close to standard multi-node training within a HPC cluster.
>     - In contrast, our method and the baselines that we compare against all use standard Federated Training technique with communications made only after $T$ steps of local training on local data. This arrangement makes the training task more challenging, simulates real-world scenarios more accurately, and substantially reduces communication requirements, leading to differing results. In essence, FedAvg-FT serves as the standard baseline for the non-IID setting we select.
>      - We also note that SSFL outperforms our standard *dense* baseline FedAvg-FT (and FedAvg) in the non-IID setup at the lower sparsity level of $50\%$. This phenomena agrees with those reported in single node sparse training, where sparse models in lower sparsity range outperformed dense baselines [9].
>
> 2. **SSFL and selected baselines tested on the harder non-IID Assumption:**
>     - An even more significant difference is that all the methods [1,2,3] use the general CIFAR10 dataset assuming IID condition, while our work and all the baselines we report were tested on the CIFAR10 non-IID split dataset where the class allocation is done according to the Dirichlet distribution with $Dir(\alpha)$ [5]. Moreover, they also do not prune the whole network and leave the first and last layer intact heuristically.
>     - Note, that this is a considerably harder task compared to the standard Cifar10 task by itself without sparsity considerations. As a further validation of this argument, we note that the last reference shared by the reviewer FedPM [4] which uses the non-IID Cifar10 split also report a accuracy drop in general and compared to us, a lower accuracy as well (57.30\% at optimal sparsity vs our 84.18% at 95% sparsity). Although, we believe there are more reasons for this drop in accuracy in FedPM that we explain later.

---

> ### Author Response · Authors · 2023-11-21
> **Response to the review of mEiz (part 2/2)**
>
> -------
> ### Comparison with FedPM, topk and random masking in the non-IID setting
>
> After noting the differences between the works suggested by the reviewer and SSFL and the baselines, taking into the reviewer's recommendation we conducted experiments with top-k, random-k and also compare FedPM (which used non-IID) We present the results from our experiments with FedPM, top-k and random masking. We note:
> - We report comparison with FedPM on Resnet-18 trained on the Cifar10 non-IID dataset with $Dir(\alpha = 0.3)$.
> - FedPM does not train the model weights and instead tries to find a super-mask and optimally determines its sparsity. This is effectively in line with works on super-masks [6]. Moreover, they use a heuristic for splitting the data in a non-IID manner.
> - The top-k presented in the recommended works was implemented in relation to gradient sparsification in DSGD setting. For our training paradigm where the sparse weights are shared every $T$ local training steps, for the top-k experiment we resorted to choosing the top-k weights at the end of each communication round to share among the clients and aggregated them.
> - We also note that at the end of training, SSFL and DisPFL ends up with sparse local models with potential for faster inference in contrast to top-k or random communication where the models remain dense.
>
>
>     | Methods |  sparsity 50% | sparsity 70%  | sparsity 80% | sparsity 90% |
>     |--------|--------|--------|--------|--------|
>     | **SSFL**  | **88.29** | **87.57** | **87.43** | **84.58** |
>     | **Random mask** | 41.61 | 39.80 | 19.70 | 10.60 |
>     | **Top-k**  | 53.33 | 52.33 | - | - |
>     | **FedPM**   | 57.30 (optimal sparsity) | N/A | N/A | N/A |
>
>     **Table:** *Accuracy at given sparsity levels. We note that FedPM optimally chooses its sparsity level and does not allow sparsity sweeps hence other sparsity levels are not available N/A. At this stage we present Top-k for 50% and 70% as the runs are computationally demanding and are still computing, however we do not expect higher accuracy at higher sparsity levels*.
>
>
> ### Local Masks
> > Would it be possible to let each client find their own mask rather than aggregating them to have one global mask for all the clients. This way, different portions of the model could be trained by different clients at the cost of losing the communication gain from server to client communication -- which is typically not the main bottleneck compared to client to server communication especially given that the current approach results in considerable accuracy loss.
>
> - We thank the reviewer for this insightful suggestion. This is indeed a very interesting idea and is a natural progression from our work However, we note that we briefly explored local masking and training but it did not result in competitive performance ($70.22$% accuracy at $50$% sparsity level on non-IID cifar10). We hypothesize that this might be due to the complex way in how completely non-overlapping masks and weights interacted and updated with each other after aggregation along with random sampling of clients. It might be the case that the local models would drift far away in the restricted (sparse) solution space and their aggregation resulting in subpar models. This would indeed be a an interesting avenue for future research.
>
> Finally, we appreciate the reviewer's careful comments and suggestions and hope that the reviewer's concerns were addressed. Should there be any additional questions or points of discussion, we are happy to discuss further.
>
> --------
> ### References
>
> [1] Lin, Yujun, et al. "Deep Gradient Compression: Reducing the Communication Bandwidth for Distributed Training." International Conference on Learning Representations. 2018.
>
> [2] Konečný, Jakub, et al. "Federated learning: Strategies for improving communication efficiency." arXiv preprint arXiv:1610.05492 (2016).
>
> [3] Barnes, Leighton Pate, et al. "rTop-k: A statistical estimation approach to distributed SGD." IEEE Journal on Selected Areas in Information Theory 1.3 (2020): 897-907.
>
> [4] Isik, Berivan, et al. "Sparse Random Networks for Communication-Efficient Federated Learning." The Eleventh International Conference on Learning Representations. 2022.
>
> [5] Hsu, T, et al. "Measuring the effects of non-identical data distribution for federated visual classification." arXiv preprint arXiv:1909.06335, 2019.
>
> [6] Deconstructing Lottery Tickets: Zeros, Signs, and the Supermask. [NeurIPS 2019]
>
> [7] DisPFL: Towards Communication-Efficient Personalized Federated Learning via Decentralized Sparse Training. [ICML 2022]
>
> [8] Sparse Random Networks for Communication-Efficient Federated Learning. [ICLR 2023]
>
> [9] How well do sparse imagenet models transfer? [CVPR 2022]

---

> ### Comment · Reviewer_mEiz · 2023-11-23
> **response to the rebuttal**
>
> I thank the authors for the detailed response. The explanation on the local masks makes sense to me. It would be good to include this observation in the paper as well.
>
> As for the comparison with other sparse methods, I am not sure why the top-k baseline performs so poorly in the additional experimental results. Even in the federated setting with non-IID data, top-k has been shown to preserve accuracy for sparsity much higher than 50% when it's supported by error feedback. I also checked the revised manuscript and couldn't see details on the implementation of the top-k algorithm. Did the authors include error feedback?

---

> ### Author Response · Authors · 2023-11-23
> **Response to the reviewer's response to the rebuttal**
>
> > I thank the authors for the detailed response. The explanation on the local masks makes sense to me. It would be good to include this observation in the paper as well.
>
> Thank you for your comment and we believe this suggestion enhanced the exposition as well! We added a discussion about the local mask first in section 3.1 and in more detail in Appendix-A.2.
>
> > As for the comparison with other sparse methods, I am not sure why the top-k baseline performs so poorly in the additional experimental results. Even in the federated setting with non-IID data, top-k has been shown to preserve accuracy for sparsity much higher than 50% when it's supported by error feedback. ...  Did the authors include error feedback?
>
> We emphasize that upon completion of Federated Learning (FL) training, both SSFL and DisPFL [7] *yield sparse local models*, in contrast to top-k methods which produce dense models. Consequently, local models trained with these methods reap the benefits of sparsity, including accelerated inference. This output of *sparse models* was previously cited in the Algorithm; we now explicitly highlight this characteristic in the paper's contributions for added clarity.
>
> We acknowledge the existence of papers [1,3] which utilize a top-k approach for *gradient sparsification*. We have attempted to find a reference for the federated non-IID setting as mentioned by the reviewer, where top-k is used for *weight aggregation* instead of *sparse gradient* training as in [1,3,11]. However, our search that included this recent survey [12] did not yield any references featuring our specific configuration, which involves infrequent *weight averaging* every $T$ steps as opposed to *gradient sparsification*. Could the reviewer please specify the paper you are referencing that employs this setup?
>
> Please note that adding an error-feedback mechanism to our weight aggregation approach (instead of gradient sparsifcation) would constitute a novel contribution. While it is an intriguing idea, developing such a method is outside the scope of the current study. We believe that our decision to forgo explicitly implementing a variant of error-feedback for our problem setup at this stage is justified, as it keeps our approach aligned with established methods and maintains the clarity of our results.
>
> We trust this clarifies our methodology, and we are genuinely grateful for your constructive suggestions that helped enhance our research.
>
> > I also checked the revised manuscript and couldn't see details on the implementation of the top-k algorithm.
> - We uploaded a new revised manuscript with the implementation details briefly explained in section-4 of the main paper and in more detail in Appendix A.2. Moreover, the implementation can be accessed via anonymous github repo noted within the supplementary materials.
>
> We thank the reviewer for their careful comments and engagement, and we are happy to discuss further.
>
>
> -----
> ### References
> [10] Sparse Communication for Distributed Gradient Descent. [arXiv, 2017]
>
> [11] The Error-Feedback Framework: Better Rates for SGD with Delayed Gradients and Compressed Updates. [arXiv, 2021].
>
> [12] Towards Efficient Communications in Federated Learning: A Contemporary Survey [Journal of the Franklin Institute, 2023].

---

### Author Response · Authors · 2023-11-21
**Common Response**

We sincerely appreciate the comments and constructive suggestions from all the reviewers. We are glad that Reviewer mEiz considers our work to be intuitive and easy to grasp and liked the presentation, finding the paper well-written. Similarly, Reviewer 1qhQ noted the novelty of our work and appreciated the presentation, while Reviewer p421 appreciated its simplicity, efficiency, and our comprehensive literature review. We would also like to thank the reviewers for suggesting additional evaluations that have helped improve the exposition. We provide a revised manuscript and color-code the changes for the reviewers: brown (mEiz), orange (1qhQ) and purple (p421).

### List of Revisions
We list the addressed points in each individual responses and list the changes made to the revised manuscript.

**Reviewer mEiz**
- We have carefully addressed the differences between the suggested works and our own along with chosen baselines, providing a detailed explanation for the observed differences in performance. Furthermore, the related works section in the manuscript has been updated to reflect the same.
- We explain the reviewer's suggestion of local masking, conduct an experiment and add a paragraph discussing it in Appendix A.2.
- We conduct new experiments on top-k, random-k and FedPM and report the results as per the reviewer's recommendation. Updated Table-1 and Figure-2 (a) in the revised paper to reflect this.

**Reviewer 1qhQ**
- We address the reviewer's points and add discussions on the topics suggested by the reviewer in our related works (Introduction) section of the revised manuscript.

**Reviewer p421**
- We update section 3.1 and add Appendix A.1 addressing the reviewer's central point regarding the validity of the mask generation process.
- We note that sparsity sweeps requested (including $<10 \%$ density) are already present in the original manuscript Figure-2. We report sparsity/accuracy comparison with more methods in our revision, further addressing the comment.
- We conduct longer duration $R=800$ communication round and include new longer duration convergence plots in Fig-5, Appendix A.2.
- We conduct random masking experiments as the reviewer recommends and update Table-1 and Figure-2 in the paper with the results.

We address each point of the reviewers in better detail in the individual responses.

---
**Summary of new Experimental Results**

We list new experimental results on top-k, random masking, and compare with SubFedAvg and FedPM below on the non-IID CIFAR10 dataset. We note for clarity: for Random masking we choose random masks for all clients at initialization and train using those masks. DisPFL similarly chooses random masks at initialization but updates them during training. We (SSFL) use a saliency criterion to generate relevant masks at initialization and train.

| Methods     | 50% | 70% | 80% | 90% | 95\% |
|--------|--------|--------|--------|--------|--------|
| **SSFL** | **88.29** | **87.57** | **87.43** | **84.58** | **84.18** |
| **DisPFL** | 85.82 | 83.39 | 82.93 | 76.17 | 75.00 |
| **SubFedAvg** |79.70 | 84.89  | 78.18 | 78.13 | 76.77 |
| **Random mask** | 41.61 | 39.80 | 19.70 | 10.60 | N/A |
| **FedPM** | 57.30 (optimal) | N/A | N/A | N/A | N/A |
| **Top-k** | 53.36 | 52.33 | - | - | - |

**Table:** *Accuracy at given sparsity levels. We note that FedPM optimally chooses its sparsity level and does not allow sparsity control, hence other sparsity levels are not available N/A. At this stage we present Top-k for 50% and 70%, as the runs are computationally demanding and are still computing; however we do not expect higher accuracy at higher sparsity levels. We will include completed sweeps in our final version.*

As suggested by reviewer mEiz we implement a variant of top-k in our training paradigm and report results we obtained. We report random-k masking (recommended by reviewer mEiz and p421), FedPM (recommended by mEiz) and also sweep SubFedAvg over sparsity range and note the efficacy of SSFL over other sparse methods.

---

### Meta-Review · Area_Chair_buXC · 2023-12-07

**Metareview:**

Summary:
The paper proposes a new communication-efficient federated learning (FL) framework, SSFL, that initially finds a binary sparsity mask using the non-iid dataset across clients and then trains a sparse network extracted by this mask during the FL stage. This reduces the communication cost in the FL stage as each client trains and communicates a sparse model at every round.

Strengths:
+ The proposed method is intuitively easy to grasp, and the paper is well-written. The empirical results suggest an improvement over the selected baselines.
+ The presentation of this paper is good.
+ This paper introduces a simple and efficient approach to subnetwork extraction.

Weaknesses:
- A complete experimental justification and ablation study is missing. See initial reviews.
- Comparing SSFL to alternative sparse methods (Top-k[1], Random-k[2], or rTop-k[3]) would provide a more complete picture to the readers.
- It is not clear whether the system will find a reasonable mask per client at the beginning of the execution.
- The design of the system follows that of FedAvg (minor contribution).
- The experiments only show the result on a sparsity level of 50%, which is not convincing. Reviewers would like to see the performance results when the sparsity level is 10% or smaller.

**Justification For Why Not Higher Score:**

For the reasons above

**Justification For Why Not Lower Score:**

N/A

---

### Decision · Program_Chairs · 2024-01-16

Reject